# Plasma miR-486-5p Expression Is Upregulated in Atrial Fibrillation Patients with Broader Low-Voltage Areas

**DOI:** 10.3390/ijms242015248

**Published:** 2023-10-17

**Authors:** María Cebro-Márquez, Moisés Rodríguez-Mañero, Valentina Serrano-Cruz, Marta E. Vilar-Sánchez, Laila González-Melchor, Javier García-Seara, José Luis Martínez-Sande, Alana Aragón-Herrera, María Amparo Martínez-Monzonís, José Ramón González-Juanatey, Ricardo Lage, Isabel Moscoso

**Affiliations:** 1Cardiology Group, Centre for Research in Molecular Medicine and Chronic Diseases (CIMUS), Instituto de Investigación Sanitaria de Santiago de Compostela (IDIS), Universidade de Santiago de Compostela, 15782 Santiago de Compostela, Spain; maria.cebro.marquez@usc.es (M.C.-M.); valentina.serrano@rai.usc.gal (V.S.-C.); marta.vilar.sanchez@rai.usc.es (M.E.V.-S.); 2Department of Cardiology and Coronary Unit, Instituto de Investigación Sanitaria de Santiago de Compostela (IDIS), Complexo Hospitalario Universitario de Santiago de Compostela, 15706 Santiago de Compostela, Spain; moises.rodriguez.manero@sergas.es (M.R.-M.); laila.gonzalez.melchor@sergas.es (L.G.-M.); javier.garcia.seara@sergas.es (J.G.-S.); jose.luis.martinez.sande@sergas.es (J.L.M.-S.); alana.aragon.herrera@sergas.es (A.A.-H.); maria.amparo.martinez.monzonis@sergas.es (M.A.M.-M.); jose.ramon.gonzalez.juanatey@sergas.es (J.R.G.-J.); 3Centro de Investigación Biomédica en Red de Enfermedades Cardiovasculares (CIBERCV), 28029 Madrid, Spain; 4Cellular and Molecular Cardiology Research Unit, Instituto de Investigación Sanitaria de Santiago de Compostela (IDIS), Complexo Hospitalario Universitario de Santiago de Compostela, 15706 Santiago de Compostela, Spain; 5Department of Biochemistry and Molecular Biology, Faculty of Medicine, Instituto de Investigación Sanitaria de Santiago de Compostela (IDIS), University of Santiago de Compostela, 15782 Santiago de Compostela, Spain

**Keywords:** atrial fibrillation, microRNAs, low-voltage areas, biomarkers, miR-486-5p

## Abstract

Atrial fibrillation (AF) is the most common arrhythmia worldwide, affecting 1% of the population over 60 years old. The incidence and prevalence of AF are increasing globally, representing a relevant health problem, suggesting that more advanced strategies for predicting risk stage are highly needed. miRNAs mediate several processes involved in AF. Our aim was to identify miRNAs with a prognostic value as biomarkers in patients referred for AF ablation and its association with LVA extent, based on low-voltage area (LVA) maps. In this study, we recruited 44 AF patients referred for catheter ablation. We measured the expression of 84 miRNAs in plasma from peripheral blood in 3 different groups based on LVA extent. Expression analysis showed that miR-486-5p was significantly increased in patients with broader LVA (4-fold, *p* = 0.0002; 5-fold, *p* = 0.0001). Receiver operating characteristic curve analysis showed that miR-486-5p expression could predict atrium LVA (AUC, 0.8958; *p* = 0.0015). Also, miR-486-5p plasma levels were associated with AF-type (AUC, 0.7137; *p* = 0.0453). In addition, miR-486-5p expression was positively correlated with LVA percentage, left atrial (LA) area, and LA volume (r = 0.322, *p* = 0.037; r = 0.372, *p* = 0.015; r = 0.319, *p* = 0.045, respectively). These findings suggest that miR-486-5p expression might have prognostic significance in LVA extent in patients with AF.

## 1. Introduction

Atrial fibrillation (AF) is the most common heart rhythm disorder. It affects approximately 1% of the population over 60 years of age and can affect 10% of the population over 75 years of age; thus, AF incidence increases with age. The incidence and prevalence of AF are increasing globally, representing a relevant health problem since it highly increases the risk of death in those who suffer AF; in addition, AF contributes to a significant increase in health costs [1]. In recent years, it has been reported that ablation treatment success in paroxysmal AF patients is about 80% after 5 years, decreasing to 60% after 10 years [2]. In persistent AF, ablation success is about 25% after a single procedure and 68% after multiple procedures, after approximately 7 years [2]. Atrial structural remodeling has been recognized to contribute to the perpetuation of AF [3]. Atrial structural remodeling is related to changes in cellular organelles, cells, and tissue. Characterized via atrial stretch and atrial dilation, both processes increase arrhythmias by increasing cellular hypertrophy, fibroblast proliferation, and tissue fibrosis [4,5]. Nevertheless, AF recurrence after ablation is highly correlated with previous atrial remodeling [5]. Atrial fibrosis can possibly be estimated via an electroanatomic mapping (EAM) system [6], combining electrophysiological data with anatomical information for the construction of 3D endocardial maps [7,8], thereby becoming an essential tool to assess the underlying substrate for the presence of low-voltage areas (LVAs) at the time of the ablation [9]. LVAs have been described both in paroxysmal and persistent AF, hypothesizing that LVAs are not necessarily related to AF duration [10]. Therefore, it is important to identify new indicators that allow the establishment of new risk scores and AF management beyond the type of AF since underlying mechanisms are not fully understood [11]. Recent research has suggested that microRNAs (miRNAs) might play a role in the pathogenesis of AF [12]. Current data suggest that the increased characterization and correlation of fibrosis degree, DNA, and circulating miRNA profiles might serve to establish a predictive risk score [11,13]. The identification of non-invasive biomarkers that allow for establishing the fibrosis degree in the atrium and that help to determine ablation procedures and the treatment of patients would provide a great benefit in clinical practice; moreover, they could potentially represent treatment targets [12,14,15,16].

In this study, we aimed to identify miRNAs with a prognostic value as biomarkers in consecutive patients referred for AF ablation. Additionally, we evaluated if miRNA expression is also associated with LVA extent (used as a surrogate marker for atrial fibrosis), based on LVA maps, and the possible mechanisms that can contribute to AF pathophysiology.

## 2. Results

### 2.1. Characteristics of AF Cohort

The demographic, clinical, and treatment characteristics of the participants are shown in Table 1. A total of 44 AF patients were included in the study. There was no significant difference between groups in age, gender, BMI, diabetes, hypertension, and smoking. The AF groups were based on the extent of LVA (used as a surrogate marker for atrial fibrosis from LVA maps), and 32% of patients did not present any LVA. Only a minority of patients (<10%) underwent AF mapping since the cardioversion did not restore sinus rhythm. There were statistical differences in angiotensin-receptor blocker and DHP Ca channel blocker treatments, triglycerides levels, and echocardiographic parameters, such as LA volume, left ventricular end-diastolic volume, left ventricular end-systolic volume, epicardial fat tissue volume, and heart rate; no differences were found in antiarrhythmic drug therapy (Table 1).

### 2.2. Different Expression of microRNAs with LVA in AF Patients

Plasma expression profiles for a panel of 84 miRNAs showed that miRNAs levels were differentially regulated according to LVA percentage groups in peripheral blood. The expression levels of miRNAs—hsa-let-7b-5p, hsa-miR-320a, and hsa-miR-486-5p—were increased in patients with higher LVA percentages (Figure 1a).

The area under curve data showed that only hsa-miR-486-5p may be a good predictor of LVA percentage in patients in Stages 1 and 3 in plasma from peripheral blood (Figure 1b and Appendix A).

### 2.3. Different Expression of microRNAs with AF Type

Regarding miRNA profile based on AF type, first, we found an association between LVA percentage and AF type (Figure 2a,b), resulting in long-standing persistent (LS persistent) patients with broader LVA.

Our data also showed that hsa-let-7b-5p and hsa-miR-486-5p were differently expressed depending on AF type (Figure 3a). The area under curve data showed that hsa-miR-486-5p may be a predictor of AF type for persistent vs. long-standing persistent patients in plasma from peripheral blood (Figure 3b and Appendix A).

### 2.4. MicroRNA Correlation with Clinical Data

Correlation analysis revealed that miR-486-5p expression was positively correlated with LVA percentage (Figure 4a), LA area (Figure 4b), and LA volume (Figure 4c).

### 2.5. KEGG Pathways and Prediction Targets

Regulated miR-486-5p was predicted to target multiple genes involved in several pathways related to cardiovascular diseases. The KEGG analysis included pathways that are related to the AGE-RAGE signaling pathway in diabetic complications; cellular senescence; valine, leucine, and isoleucine degradation; propanoate metabolism; FoxO; p53; TGF-beta; AMPK; Insulin; signaling pathways regulating the pluripotency of stem cells; mTOR; and focal adhesion signaling pathways (Figure 5).

## 3. Discussion

Atrial fibrosis is a common pathological feature of AF, characterized by the excessive deposition of extracellular matrix proteins, such as collagen and fibronectin, within the atrial tissue; this fibrosis can contribute to the initiation and maintenance of AF by promoting conduction abnormalities, electrical remodeling, and structural changes in the atrial tissue [4]. Our study tries to elucidate whether the expression profiles of miRNAs could have prognostic significance in LVA extent in patients with AF. The DECAAF study investigated the use of delayed-enhancement cardiac magnetic resonance imaging (DE-CMR) to predict the recurrence of AF after catheter ablation. The study found that patients with high levels of fibrosis in the LA, as detected by DE-CMR, were more likely to experience recurrent AF after catheter ablation compared to patients with low levels of fibrosis [17]. Based on the DECAAF study, we have established three groups of patients taking into account the extent of LVA, as determined via EAM (Stages 1–3). After miRNA circulating profile analysis, we found that miR-486-5p is overexpressed in patients with higher LVA extent. Receiver operating characteristic curve data show that only hsa-miR-486-5p is a good predictor of LVA extent and is a good biomarker for prognostic patient stratification. Furthermore, LVA percentage, LA area, and LA volume are positively correlated with miR-486-5p expression. Regarding AF type, miR-486-5p is also overexpressed in long-standing persistent AF patients compared to paroxysmal or persistent AF patients; also, receiver operating characteristic curve analysis demonstrates that miR-486-5p is a good predictor of AF type. In this sense, another study found a significant correlation between miR-21 serum concentration and the extent of LVA detected in the LA, but only persistent AF patients were included, and less extreme LVAs were selected to stratify the groups of AF patients [13].

It has been previously reported that miR-486-5p also plays a role in regulating the expression of genes that are involved in electrical conduction and structural heart remodeling [18]. Several studies have shown that miR-486-5p is upregulated in patients with AF [19,20,21] and arrhythmias [22] and is also associated with a greater risk of developing AF [23]. Besides its role in AF development, miR-486-5p has also been implicated in the formation of scar tissue in the heart. Several studies have shown that miR-486-5p expression is upregulated in patients with cardiac fibrosis, and that higher levels of miR-486-5p are associated with a greater degree of fibrosis [24,25]. This suggests that miR-486-5p may be involved in the regulation of fibrotic processes in the heart. Mun et al. [21] found that miR-486 is highly expressed in patients with persistent AF compared with supraventricular tachycardia patients. miR-486-5p association with arrhythmias was demonstrated by Li et al. [22], who showed that it is overexpressed in patients who experienced arrhythmias, and it is related to sinoatrial node function depression. Other studies found that in LA tissue, miR-486-5p is upregulated in AF patients compared to sinus rhythm controls, and this is related to the fact that it induces the accumulation of superoxide anion, DNA damage, and reduced cell proliferation, contributing to a senescent phenotype in human fibroblasts [20]. Additionally, Wang et al. [19] found that miR-486-5p is elevated in the LA in patients with non-valvular paroxysmal AF. Thus, it has been demonstrated that persistent patients showed more fibrosis in the LA than paroxysmal patients, presenting 2- to 3-fold more fibrosis than non-AF patients, showing a significant correlation between fibrosis and AF severity, in which long-standing persistent patients presented more fibrosis than paroxysmal AF patients, which is also consistent with our results [26]. Furthermore, in studies performed on explanted human atria, both MRI and histology studies showed significant fibrosis in both atria in AF patients; these data suggest that fibrosis highly correlates with AF and also plays a significant role in AF maintenance [27]. However, our results showed that miR-486-5p expression correlates with LVA percentage, LA area, and LA volume. Particularly, patients with persistent AF exhibit larger LA diameters than supraventricular tachycardia patients, suggesting that miR-486 expression could present LA enlargement [21].

Targeting analysis showed that dysregulated genes, targeted by miR-486-5p, could be involved in AF maintenance and fibrosis, supporting our results (Figure 6). miR-486 regulates Pim-1, a kinase protein that phosphorylates cTnI, with both colocalized in cardiomyocytes. Pim-1 phosphorylates cTnI and, in myocyte contraction, when cTnI is phosphorylated, misses its sensibility for calcium, producing muscle fiber relaxation [28]. Pim-1 may be downregulated in AF patients through miR-485-5p overexpression, leading to the maintenance of the contraction when cTnI phosphorylation is reduced [29]. However, according to the mechanisms of AF maintenance, miR-485-5p targets PCCA. In a PCCA knockout murine model, cardiac dysfunction was associated with lower systolic Ca^2+^ release, impairment in the sarcoplasmic reticulum Ca^2+^ load, and decreased Ca^2+^ re-uptake via SERCA2a. These abnormalities are common in atrial cardiomyocytes of AF patients. In addition, it was reported that mutations in the PCCA gene were related to long QT syndrome, which is often associated with FA [30]. The CADM1 target gene may also be implicated in AF fibrosis via miR-486 overexpression. STAT3 is increased in activated fibroblast and fibrosis tissue. CADM1 expression is decreased in cardiac fibrosis tissue and fibroblast and may regulate STAT3 controlling cellular proliferation and, therefore, cardiac fibrosis development [31].

In summary, miR-486-5p and its targeted genes have been implicated in the pathophysiology of AF and the formation of scar tissue in the heart. Higher levels of miR-486-5p are associated with a greater risk of developing AF and a greater degree of cardiac fibrosis, which can contribute to impaired cardiac function.

### Limitations

This study has some limitations that need to be taken into account at the time of data interpretation. Firstly, and very importantly, this study was conducted with a small sample size in a single center. Thus, the expression of miR-486-5p needs to be validated in a larger sample size. One of the limitations of our study is that it is a non-randomized retrospective study. In addition, 16% of our patients had already undergone pulmonary vein ablation. Patients with radiofrequency lesions beyond the pulmonary veins were not included in the analysis because they may interfere with the level of myopathy determined via electroanatomic voltage mapping. Thus, the quantification of the LVA percentage was performed outside the pulmonary veins, so this does not influence the conclusions of this study. Spontaneous low-voltage regions are a surrogate marker for atrial fibrosis, but histological validation is missing. The spatial distribution and extent of LVZ depend largely on spontaneous rhythm and the site and frequency of atrial pacing, as well as the mapping catheter and interelectrode distance. However, all our patients were mapped in spontaneous rhythm without pacing, using catheters with the same electrode size, interelectrode distance, and automatic acquisition setting validated in our previous studies. Multipolar catheters may be prone to suboptimal contact in several LA regions. The definition of the total LA surface area may not be in line with previous studies since, for the present analysis, it was defined as the LA body area without the PV antrum regions, LA appendage orifice, and mitral valve. Although it is our belief that it should not alter the conclusions of this study (regarding the amount of LVA), our conclusions should be interpreted under this definition. To conclude, the temporal pattern and intermittent ECG monitoring determined the AF burden, although this did not really correspond to long-term ECG monitoring.

## 4. Materials and Methods

### 4.1. Patients

We included consecutive AF patients referred for pulmonary vein ablation despite having optimal pharmacological therapy at the University Clinical Hospital of Santiago de Compostela. The exclusion criteria were age under 18 years, any latent infectious condition, and pregnancy, and there was no history of malignant chronic kidney disease or osteoarthritis present. Patients with radiofrequency lesions beyond the pulmonary veins were not included in the analysis because they might interfere with the level of myopathy determined via electroanatomic voltage mapping. Due to the fact that previous radiofrequency ablation beyond the pulmonary veins generates areas of low voltage at the time of the EAM, patients with previous AF ablation outside the pulmonary veins were excluded from the analysis. Also, patients with paroxysmal, persistent, and long-standing persistent were included. Antiarrhythmic drug therapy (ADT) was continued during the blanking period (defined as 3 months after ablation), and after this period, only ADT was restarted in case of recurrence. This study complies with the Declaration of Helsinki and was approved by the Clinical Research Ethics Committee of Galicia (MRM-miRAF-2017-01). All of the patients signed informed consent.

### 4.2. AF Assessment

AF was classified, according to the American Heart Association and American College of Cardiology, as paroxysmal (intermittent in nature, terminating spontaneously or within 7 days of treatment); persistent (failure to terminate in 7 days); long-standing persistent (lasting for more than 12 months) [32].

In all patients, AF was recorded using a 12-lead electrocardiogram (ECG) within the 6-month period before ablation. Computed tomography (CT) or magnetic resonance imaging (MRI) was routinely performed and used to guide the manipulation of the catheter at the time of the procedure.

### 4.3. Surgery Intervention and Sample Collection

As previously described by López-Canoa et al. [33], patients were submitted to a night of fasting. Firstly, only before the ablation procedure, a peripheral blood sample was obtained using an 18-G butterfly cannula with a two-syringe technique from an ante-cubital vein; the first 5 mL was discarded, and the second 5 mL was collected. Blood samples from peripheral blood were collected in EDTA tubes. After collection, blood samples were placed on ice. Blood samples were centrifuged, and the plasma or whole blood was stored at −80 °C until the subsequent test.

### 4.4. Ablation, Acquisition of Electroanatomical Voltage Maps, and Patient Follow-Up

All procedures were performed under general anesthesia or conscious sedation with blood pressure monitoring. Trans-esophageal echocardiography or CT-angiography was performed in all patients to rule out the presence of left atrial thrombus before ablation. If already present, OACs were not interrupted before the procedure. Vitamin K antagonists were continued with a target INR between 2 and 3. Direct oral anticoagulants (DOACs) were discontinued on the day of the procedure and resumed the same day. After groin puncture, intravenous heparin was administered to maintain an activated clotting time between 300 and 350 s throughout the whole procedure. Concerning the ablation, irrigated tip ablation catheter with contact force sensing technology was systematically employed. RF lesions were placed in temperature-limited power control mode (30–40 watts at the posterior and 35–45 watts at the anterior wall). The ablation was guided via automatic ablation annotation and minimum force–time integral and, later, minimum ablation index values, local electrogram attenuation, and impedance changes. Antral or wide-antral PVI was performed at operator’s discretion, but ostial ablation was avoided. Entry and exit blocks of the PVs were assessed with or without intravenous adenosine, and if needed, touch-up applications were applied at the gap sites to achieve block. Point-by-point pulmonary vein isolation was performed in all patients using contact force sensing technology (SmartTouch, Biosense Inc., Diamond Bar, CA, USA) following high-density bipolar voltage mapping. The LVA percentage was defined via the size of the LA fibrotic area, derived from a bipolar voltage map created simultaneously with LA surface reconstruction, guided by an EAM system (CARTO3, Biosense Webster, Diamond Bar, CA, USA) using a multipolar mapping catheter (PentaRay, Biosense Webster, Diamond Bar, CA, USA). Patients in AF rhythm at the start of the procedure systematically underwent electrical cardioversion in the electrophysiology laboratory. For those patients who relapsed after at least two cardioversions, EAM was performed in AF, adjusting the cut-off [8]. Adequate quality of the acquired voltage points was established according to CONFIDENSE module after respiratory compensation. This is a continuous mapping software with automated data acquisition when set criteria are met, namely, (1) tissue proximity indication; (2) map consistency (which means a reasonable time of activation as compared to contiguous points; if it does not fulfill criteria, the point is reexamined); (3) position stability filter (4 mm); (4) cycle length stability (keeps data collected within a range of predefined cycle lengths, within 10% of the average). Wavefront annotation was systematically activated. A minimum number of points was requested (>1000), and the density fill threshold remained constant at ≤5 mm. Contiguous areas of bipolar voltage <0.5 mV were considered LVA in sinus rhythm [33,34]. Total LA surface area was defined as the LA body area without the PV antrum regions, LA appendage orifice, and mitral valve. Medians of the total LA surface area and area of each predefined region were measured offline on the three-dimensional reconstructed LA model. Median values of LVA were set in relation to the surface area of each region and the entire LA. Carto-3 built-in software was used to calculate the percentage of LVAs from the LA surface. Maps were color-coded from grey (<0.5 mV, substantial LVAs) to purple (>1.5 mV, normal voltage). Patients were assigned to 1 of 3 groups (Stages 1–3) based on the volumetric percentage of LA wall enhancement [5,35] and more extreme LVAs: Stage 1 (<10% of atrial wall); Stage 2 (≥10%–<50% of atrial wall); and Stage 3 (≥50% of atrial wall).

### 4.5. RNA Extraction and miRNA Quantification

Sequences of 84 different predesigned mature miRNAs (listed in Appendix A) were detected using a Human Cardiovascular Disease miScript miRNA PCR Array (MIHS-113Z, Qiagen, Hilden, Germany), as previously described, containing a miRNA sequence from *C. elegans* as an exogenous normalizer (spike-in cel-miR-39) [36]. All cDNA steps and PCR setup were performed via a QuantStudio™ 5 Flex Real-Time PCR System (Applied-Biosystems, Carlsbad, CA, USA). The PCR cycling was performed according to the manufacturer’s protocol. Briefly, only miRNAs with Ct values < 30 in all samples were considered. miRNA normalized expressions were represented by ∆Ct, calculated by subtracting the global geometric mean signal from individual miRNA Ct values. The 2−∆∆Ct method was used to calculate miRNAs’ fold change.

### 4.6. Bioinformatics Analysis for miRNA Target Genes and Biological Pathways

miRNA-targeted genes were retrieved from the miRWalk 3.0 database (http://mirwalk.umm.uni-heidelberg.de/ (accessed 20 February 2023)). EnrichR was used for GO terms and KEGG pathway enrichment analyses (https://maayanlab.cloud/Enrichr/ (accessed on 20 February 2023)). MiRNA–genes–pathways networks were visualized with Cytoscape software 3.9.1 (http://cytoscape.org/ (accessed on 28 February 2023)). In silico analyses were performed to fully understand the functional role of differentially regulated miRNAs.

### 4.7. Statistical Analysis

The Shapiro–Wilk test was performed to test the normality of distribution. The Mann–Whitney test, Fisher’s test, and ANOVA, followed by Tukey’s post hoc test, Spearman’s correlation, and the area under receiver operating characteristic curve (AUC) analysis, were performed using GraphPad Prism 9 (GraphPad Software Inc., San Diego, CA, USA). Numerical data were presented as mean and standard deviation (SD), interquartile range (IQR), or standard error of the mean (SEM). In all analyses, a two-tailed *p* < 0.05 was considered to be significant.

## 5. Conclusions

Thus, miR-486-5p may be involved in the pathogenesis of AF and LVA extent, and its dysregulation could contribute to the development and progression of these conditions. Further research is needed to fully elucidate the role of miR-486-5p in the pathogenesis of AF and its potential as a biomarker or therapeutic target for the treatment of AF and associated LVA extent.

## Figures and Tables

**Figure 1 ijms-24-15248-f001:**
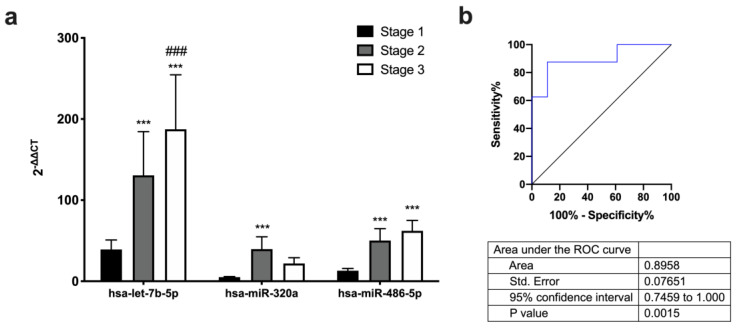
(**a**) Differential expression levels of miRNAs in AF patients between groups in plasma from peripheral blood. (**b**) Predictive capacity of LVA stage. Receiver operating characteristic curves comparing sensitivity and specificity of plasma from peripheral blood differentially expressed miR-486-5p in predicting LVA percentage. Data are presented as mean ± S.E.M. *** *p* < 0.001 vs. Stage 1 and ^###^
*p* < 0.01 vs. Stage 2.

**Figure 2 ijms-24-15248-f002:**
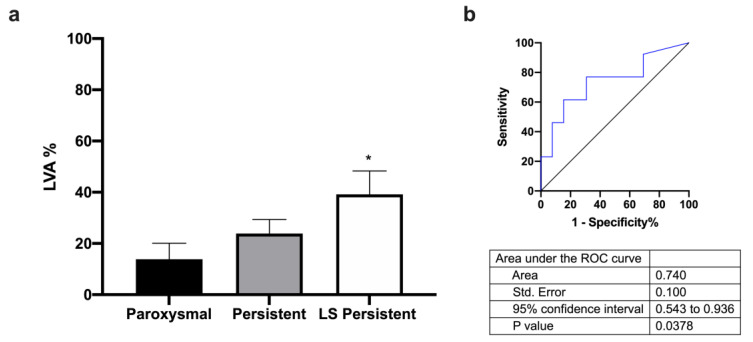
Association between LVA percentage and AF type. (**a**) LVA percentage in paroxysmal, persistent, and LS persistent patients. (**b**) Receiver operating characteristic curve comparing sensitivity and specificity of LVA in paroxysmal and LS persistent patients. Data are presented as mean ± S.E.M. * *p*< 0.05 vs. paroxysmal.

**Figure 3 ijms-24-15248-f003:**
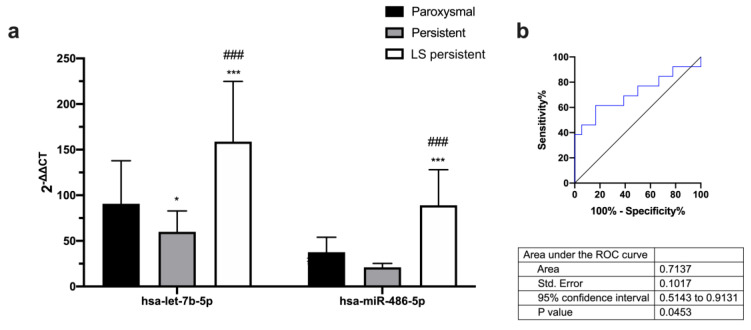
(**a**) Differential expression of miRNAs in paroxysmal, persistent, and long-standing persistent AF patients in plasma from peripheral blood. (**b**) Predictive capacity of AF type. Receiver operating characteristic (ROC) curves comparing sensitivity and specificity of hsa-miR-486-5p differentially expressed in persistent and long-standing persistent patients in plasma from peripheral blood. Data are presented as mean ± S.E.M. * *p* < 0.05, *** *p* < 0.001 vs. paroxysmal and, ^###^ *p* < 0.001 vs. persistent.

**Figure 4 ijms-24-15248-f004:**
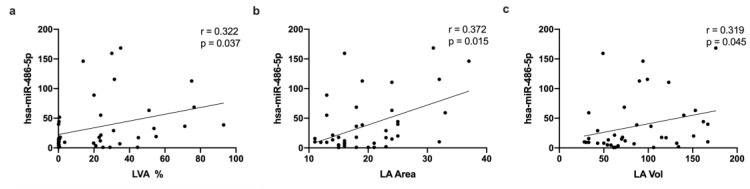
Correlation of has-miR-486-5p expression with echocardiography data. (**a**) Correlation with LVA percentage. (**b**) Correlation with LA area. (**c**) Correlation with LA volume.

**Figure 5 ijms-24-15248-f005:**
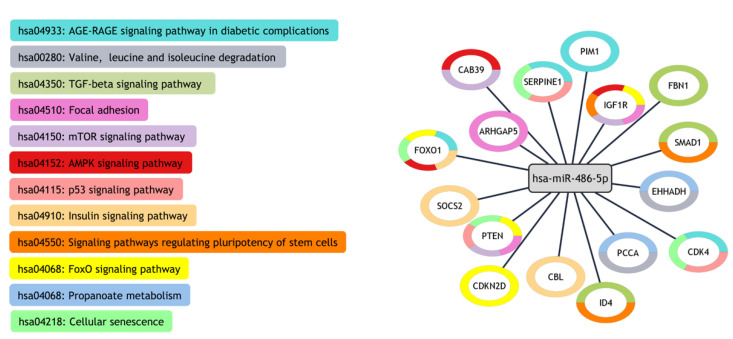
miRNA–genes–pathways network denoting the relationships between miR-486-5p, validated target genes, and those related to KEGG pathways, created in Cytoscape.

**Figure 6 ijms-24-15248-f006:**
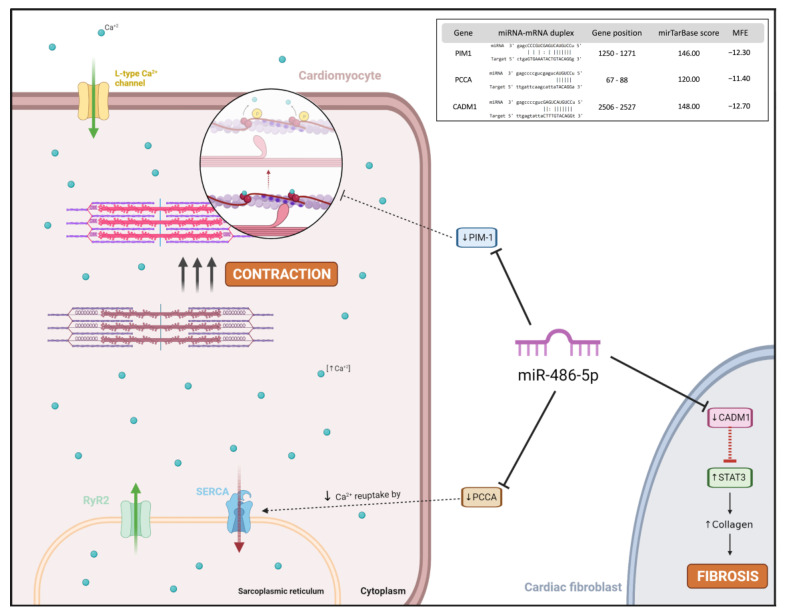
Overview of miR-486-5p target genes (miRTarBase) and possible pathological effects on AF fibrosis progression. Created with BioRender. MFE: minimum free energies.

**Table 1 ijms-24-15248-t001:** Clinical parameters. Data are presented as mean + IQR or SD, % (*n*).

Parameter	Total	All Patients		
	*n* = 44	Stage 1 (*n* = 18)	Stage 2 (*n* = 18)	Stage 3 (*n* = 8)
Age (years)	58.6 (62.5–54.8)	56.3 (60.4–52.1)	58.7 (62.9–54.5)	66.8 (73.6–59.9)
Male	31 (70.5%)	14 (73.7%)	13 (72.2%)	4 (50%)
BMI	29.2 (31.3–28.2)	29.6 (32.2–27.1)	29.8 (31.9–27.7)	30.0 (35.7–24.3)
**Pre-existing Conditions**				
Hypertension	25 (56.8%)	8 (44.4%)	11 (61.1%)	6 (75%)
Diabetes	7 (15.9%)	1 (5.5%)	3 (16.6%)	3 (37.5%)
Smoking	14 (31.8%)	6 (33.3%)	6 (33.3%)	2 (33.3%)
LVA %	24.7 (32.9–16.5)	0.4 (0.9–0)	28.3 (32.4–24.2) ***	71.1 (85.6–56.7) ***^,###^
Statines	21 (47.7%)	8 (44.4%)	7 (38.8%)	6 (75%)
ACEi	11 (25%)	5 (27.7%)	6 (33.3%)	0 (0%)
ARB	12 (27.3%)	2 (11.1%)	4 (22.2%)	6 (75%) **^,##^
DHP Ca channel blockers	5 (11.4%)	1 (5.5%)	1 (5.5%)	3 (37.5%) *^,#^
Acenocoumarol	18 (40.9%)	6 (33.3%)	8 (44.4%)	4 (50%)
NOAG	26 (59.1%)	12 (66.6%)	10 (55.5%)	4 (50%)
Class I ADT	15 (34.1%)	9 (50%)	4 (22.2%)	2 (33.3%)
Class II ADT	34 (77.3%)	15 (83.3%)	12 (66.6%)	7 (87.5%)
Class III ADT	13 (29.6%)	3 (16.7%)	8 (44.4%)	2 (33.3%)
Class IV ADT	4 (9.1%)	1 (5.6%)	2 (11.1%)	1 (16.6%)
Cholesterol	187.1 ± 43.2	185.7 ± 33.4	189.4 ± 55.4	184.9 ± 34.1
LDLc	108.9 ± 32.5	113.5 ± 26.6	106.4 ± 43.7	103.9 ± 15.2
HDLc	53.4 (59.2–47.5)	51.5 (62.8–40.2)	52.1 (60.6–43.7)	59.9 (73.7–46.0)
TG	121.7 (137.8–105.5)	127.0 (147.0–107.0)	123.8 (157.5–90.0)	105.6 (137.7–73.6) **^,#^
**FA type**				
Paroxysmal	13 (29.5%)	8 (44.4%)	4 (22.2%)	1 (16.6%)
Persistent	18 (40.9%)	7 (38.8%)	8 (44.4%)	3 (37.5%)
Long-standing persistent	13 (29.5%)	3 (16.6%)	6 (33.3%)	4 (50%)
**Echocardiographic Parameters**				
LVEF (%)	59.8 (62.4–55.9)	59.8 (64.7–55.0)	58.3 (64.8–52.4)	59.8 (66.7–55.1)
LA Area (cm^2^)	18.7 (21.3–17.7)	18.2 (21.4–15.0)	19.7 (22.6–16.7)	22.0 (26.2–17.8)
LA Vol (mL)	86.5 (104.2–76.4)	83.7 (110.5–56.9)	89.3 (109.8–68.9)	108.7 (135.0–82.5) **^,#^
LVEDV (mL)	64.7 (74.4–52.7)	58.7 (72.9–44.5)	77.3 (99.1–55.4) **	43.0 (55.4–30.6) ^###^
LVESV (mL)	25.3 (31.3–19.1)	22.4 (27.9–16.9)	32.1 (46.4–17.8)	15.8 (20.9–10.8) ^#^
LVTDD (mm)	40.4 (43.3–37.6)	39.7 (43.2–36.2)	43.9 (49.3–38.5)	34.5 (42.5–26.5)
LVTSD (mm)	27.8 (30.2–25.9)	28.4 (31.9–25.0)	29.5 (33.3–25.7)	23.8 (27.3–20.4)
EAT Vol (mL)	81 (99.3–62.7)	65.2 (78.5–51.9)	101.4 (141.2–61.7) ***	78.9 (155.7–2.1) ^#^
**ECG Parameters**				
HR	74.3 (80–67.1)	78.7 (92.4–65.1)	66.8 (73.0–60.7) *	77.3 (93.2–61.3)
PR	159.1 ± 25.0	148.8 ± 26.6	164.0 ± 25.3	165.5 ± 7.8
QRS	94.7 (98.7–91.1)	92.6 (97.1–88.0)	96.3 (102.1–90.6)	96.8 (113.7–79.8)

BMI—body mass index; ACEi—angiotensin-converting enzyme inhibitor; ARB—angiotensin receptor blocker; DHP Ca channel blockers—Dihydropyridine calcium channel blockers; NOAG—new oral anticoagulants; ADT—antiarrhythmic drug therapy (Class I: flecainide, propafenone; Class II: beta-blockers; Class III: sotalol, amiodarone, dronedarone; Class IV: calcium antagonist); LDL—low-density lipoprotein cholesterol; HDL—high-density lipoprotein cholesterol, TG—triglyceride; LVEF (%)—left ventricular ejection fraction; LA Area—left atrium area; LA Vol—left atrium volume; LVEDV—left ventricular end-diastolic volume; LVESV—left ventricular end-systolic volume; LVTDD—left ventricle telediastolic diameter; LVTSD—left ventricle telesystolic diameter; EAT Vol—epicardial fat tissue volume; HR—heart rate; PR—PR interval; QRS—QRS duration. * *p* < 0.05, ** *p* < 0.01, *** *p* < 0.001 vs. Stage 1; ^#^ *p* < 0.05, ^##^ *p* < 0.01, ^###^
*p* < 0.001 vs. Stage 2.

## Data Availability

The data presented in this study are available upon request from the corresponding authors.

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
