# Peer review of "Plasma miR-486-5p Expression Is Upregulated in Atrial Fibrillation Patients with Broader Low-Voltage Areas"

_ijms, 2023, doi:10.3390/ijms242015248_

Round 1

Reviewer 1 Report

In the present study, Cebro-Márquez et al. sought to identify miRNAs with a prognostic value as biomarkers in patients referred for AF ablation and its association with fibrosis extent. The authors suggest that miR-486-5p expression might have prognostic significance in the extent of low voltage areas in AF patients.

1. In the Abstract, please proof your results with numeric values including p values instead of only describing the results. 

2. Line 79: Please name that there was no difference “between groups” in the mentioned parameters.

3. Lines 84 + 263: Please stick to the past (in terms of grammar) when reporting your results. 

4. Table 1: Please not only give p values for multiple comparisons between 2 groups each, but also the p value for a comparison of all 3 groups (group 1 vs. 2 vs. 3). The stars and rhombs next to the parameters are not necessary as they duplicate the meaning of the p values to the right. Please indicate units for echocardiographic and ECG parameters.

5. How long was arrhythmia history? Were there any differences between groups?

6. Please improve the quality of Figures 3 + 4 + 6 (Table in the right upper corner).

7. Line 272: Please indicate whether you used bipolar or unipolar voltage. Were the areas of low voltage measures manually? If yes, by how many operators?

8. Were there any differences in the predictive value of miR-486-5p between patients with vs. without prior PVI? It represents a relevant limitation that some patients already underwent prior ablation.

9. It might be interesting to know whether the expression of miR-486-5 correlates with arrhythmia (AF/AT) recurrences after ablation. Are there any data on outcomes in your cohort?

Please use abbreviations consistently throughout the whole manuscript and check again carefully for grammar mistakes.

Reviewer 2 Report

In the presented manuscript, Cebro-Marquez and co-workers aimed to evaluate the  correlation between plasma expression of miRNAs and electroanatomical mapping derived low-voltage areas (LVA) among AF population undergoing pulmonary vein isolation.

 Unfortunately the study is badly designed and the manuscript is very badly written. It is the very first time when I see that the results are presented following introduction and the methodology of the study is presented following discussion. It shows the complete lack of knowledge how to write a scientific manuscript!

Major concerns

1.      Bipolar voltage mapping (BVM) has been shown to be a useful method to assess the incidence of LVA, most commonly considered a marker for the presence of atrial fibrosis. However LVA do not automatically  transfer into fibrosis. Moreover there is some extent to which the LA fibrosis may be detected or reclassified to normal when compared to other methods for detecting LA fibrosis, especially cardiac MRI.

2.      It is a well known fact that assessment of LVA burden depends on several factors such as:

·         Mapping on or off antiarrhythmic drugs

·         Rhythm  during mapping (sinus rhythm, AF, atrial pacing)

·         Density of mapping

·         Catheter resolution

·         Verification of collected points to exclude incorrectly annotated signals in the presence of atrial ectopy, uncaptured pacing, noise, ventricular and atrial farfield

·         Time relation to cardioversion

·         Time relation to PVI (before or after)

·         Applied voltage cut-off value. Voltages <0.5 mV correlate well with different degrees of LA structural defect, based on previous descriptions. However, this cut-off value has not been clearly validated.

The minority of those issues were taken into account in the methodology

3.      The authors used 2 different mapping systems (Ensite Velocity and CARTO3 ) what requires different mapping and ablation electrodes (not provided in the text beside Thermocool ST!). It is obvious that those electrodes (probably LASSO type as stated “spiral)” have different mapping resolution. Moreover  the Pentaray/Octaray or HD Grid electrodes provide the highest maping resolution exclusively.  Therefore LASSO detected LVA is nowadays out of standards

4.      LVA quantitative assessment was performed using dedicated CARTO3 software. How was this calculated with ENSITE platform? Moreover LVA was calculated in the relation to the total LA shell area. Therefore the final  shell must exclude  area beyond mitral valve annulus

5.      How many patients did not present any LVA? The basic result that I want to see is the relationship of  plasma miRNAs expression between individuals with and without LVA.

Furthermore there are many other concerns. This manuscript cannot be published in the present form.

Round 2

Reviewer 1 Report

Altogether, the comments have been properly addressed. The manuscript has substantially improved.

Reviewer 2 Report

There are no doubts that the revised version of the manuscript has improved a lot. However, there are still many caveats that not allow to publish the manuscript in its present form. Many issues were addressed in “the response to reviewers’ comments” but were not included in the main text!

Furthermore I do not support the general design of the study.

Major concerns

1.    The study population is very small (44 patients)

2.    The study cohort is presented as the mix of paroxysmal 30%, persistent 40% and long-standing persistent 30% AF. Moreover the voltage mapping was not performed at the same rhythm (some patients being in sinus rhythm and  some in AF). Furthermore mapping was performed regardless on or off antiarrhythmic drugs. Therefore the results are not reliable. Dataset should be homogenous in the terms of AF type, rhythm during mapping and antiarrhythmic strategy

3.    How the authors explain the very high level of detected LVA  68%?  This is not in line with the studies using multipolar high density-high resolution mapping. It raise a question whether voltage mapping was performed properly, especially that  the methodology of voltage mapping has been substantially changed comparing to the originally submitted manuscript.

4.    Statistical analysis: There is no information how numerical and categorical variables were presented throughout the manuscript. However, it seems that numerical data were presented as mean and standard deviation. One should remember that data with non normal distribution must be  expressed as median and interquartile range. I am pretty sure that nearly all variables among the dataset of 44 individuals were not normally distributed. It must be recalculated  

Other concerns

1.      Materials and Methods: Patients

-       no information about AF type (paroxysmal-persistent-longstanding persistent)

-       no information about  antiarrhythmic drugs

2.      Materials and Methods: Ablation, Acquisition of Electroanatomical Voltage Maps and Patient Follow-Up

-       As the authors focus on voltage mapping why information about ablation comes first? Anyway, it seems that mapping was performed following PVI what obviously not allow proper assessment of LVA (some of them could have been located within ablation encirclement). Moreover “Point-by-point radiofrequency catheter ablation” is not the same as pulmonary vein isolation.  Furthermore many ablation details are missing such as: the distance from PV ostium and ablation end-point

-       “Patients in AF rhythm at the start of the procedure systematically underwent electrical cardioversion  in the electrophysiology laboratory” There is no information about rhythm during mapping (sinus) and what a strategy was applied if DCCV failed.

-       CONFIDENSE module: provide information which parameters were applied instead of what can be used. Moreover: a) wavefront annotation is not a part of this module, b)  what does  map consistency mean?, c) position stability filter – provide a value

-       “Total LA surface area was defined as the LA body area without the PV antrum regions, LA appendage orifice, and mitral valve.” The measurement of LA area should exclude the surface area of LA appendage and left ventricular surface area (everything beyond mitral valve) but not  LAA orifice or mitral valve. Moreover  how the PV antra were defined?

-       No information about Mapping on or off antiarrhythmic drugs

3.      Results

-       Main text duplicate a great deal of information available in the tables or methods section.

-       Table 1 is very busy- consider deleting not significant data

-       Table 1: PR among not cardioverted persistent AF?

-       Provide the list of AA drugs

-       No information how many patients did not present any LVA (32% according to the response letter)

-       No information how many patients were not cardioverted into sinus rhythm

4.      The authors agreed that low-voltage region is a  surrogate marker for atrial fibrosis (added to limitations). However they still express LVA as defined equivalent for  fibrosis. Change this throughout the text.

Minor editing of English language is required

Round 3

Reviewer 2 Report

I would like to highlight that the authors should be acknowledged for the effort put into the manuscript. Many issues were intensively explained by the authors that I finally support.  However I still do not agree with some parts of the study design. Beside my previous concerns I am pretty sure that the golden standard for the assessment of low-voltage areas is to exclude individuals who underwent previous AF ablation. It significantly affects the results. It could explain why the amount of LVA in this study was higher than previously reported.

Finally, important caveats that must be corrected:

1.      Materials and Methods: Patients

- Provide information that study cohort consisted of paroxysmal, persistent and long-standing persistent AF

- Provide information that patients with previous AF ablation were not excluded (this information is provided in limitations exclusively). Moreover in the results explain in details previously employed PVI workflow

- Provide information that AA drugs were not discontinued throughout the study

-Delete the information about exact number of patients (44) included – this information should be reserved for results

2. Ablation, Acquisition of Electroanatomical Voltage Maps and Patient Follow-Up

- Replace the sentence “Point-by-point radiofrequency catheter ablation was performed in all patients using contact force sensing technology (SmartTouch, Biosense Inc.)” with Point-by-point pulmonary vein isolation was performed in all patients using contact force sensing technology (SmartTouch, Biosense Inc.) following high density bipolar voltage mapping”
